# Immunologic Role of Innate Lymphoid Cells against *Mycobacterial tuberculosis* Infection

**DOI:** 10.3390/biomedicines10112828

**Published:** 2022-11-06

**Authors:** William Narinyan, Nicole Poladian, Davit Orujyan, Areg Gargaloyan, Vishwanath Venketaraman

**Affiliations:** College of Osteopathic Medicine of the Pacific, Western University of Health Sciences, Pomona, CA 91766, USA

**Keywords:** innate lymphoid cell, ILC1, ILC2, ILC3, *Mycobacterium tuberculosis*, BCG

## Abstract

Tuberculosis (TB), caused by *Mycobacterium tuberculosis* (*M. tb*), is one of the leading causes of mortality due to respiratory tract infections worldwide. Infection by *M. tb* involves activation of a type I immune response characteristic of T helper type 1 (Th1) lymphocytes, natural killer (NK) cells, Interleukin-12 (IL-12), and interferon (IFN)-γ, all of which stimulate the activation of macrophages and robust phagocytosis in order to prevent further infectious manifestations and systemic dissemination. Recent discoveries about innate lymphoid cells (ILCs) have provided further insight about how these cells participate within the protective immune response against *M. tb* infection and help boost the type I immune response. In order to clearly understand the mechanisms of *M. tb* infection and advance the efficacy of future treatment and prevention, we must first look at the individual functions each type of immune cell plays within this process, specifically ILCs. By review of the recent literature and current evidence, our group aims to summarize the characterization of the three major groups of ILCs, including NK cells, and analyze the role that each group of ILCs play in the infectious process against *M. tb* in order to provide a more comprehensive understanding of the host immune response. Equally, previous studies have also highlighted the effects of how administration of the Bacille Calmette–Guérin (BCG) vaccine influences the cells and cytokines of the immune response against *M. tb*. Our group also aims to highlight the effects that BCG vaccine has on ILCs and how these effects provide added protection against *M. tb*.

## 1. Introduction

Innate lymphoid cells (ILCs) are a family of lymphocytes derived from common lymphoid progenitor cells that function as innate correlates to T cells in response to microbial infection, as depicted in Figure 1 [1]. Recognized only in the past decade, they have been found to be abundant at mucosal barriers, particularly within the gastrointestinal and respiratory tract and upon exposure to pathogens, cytokines, and microbial compounds, they release effector cytokines, normally released by CD4+ T helper (Th) cells, thereby priming the immune response against infection [2,3]. 

ILCs are categorized within two primary groups: cytotoxic and helper-ILC. The cytotoxic-ILC population is represented by natural killer (NK) cells of the innate immune system, specifically group 1 ILCs, featuring the ability to kill intracellular pathogens via antibody-dependent cellular cytotoxicity (ADCC) [4]. Helper-ILCs are classified based on the transcription factors that drive their differentiation, the cytokines they secrete, and their phenotypic criteria. ILC1s, which also include NK cells, act as an innate counterpart to Th1 cells and produce type 2 interferon (IFN)-γ when exposed to IL-12, IL-15, and IL-18, driving the proliferation and stimulation of macrophages and dendritic cells against intracellular pathogens [2,5]. ILC2s parallel Th2 cells, secreting IL-5, IL-9, IL-13, and amphiregulin when exposed to IL-25, IL-33, and thymic stromal lymphopoietin (TSLP). These cells are essential in the destruction of helminths, and for repairing tissue damage caused by helminths and viruses. ILC3s, considered as innate counterparts to Th17 cells, secrete IL-22 and IL-17 in response to IL-23 and IL-1β, which help maintain the integrity of the intestinal epithelial barrier and dominate the human respiratory tract [2,4,6]. Group 3 ILCs also encompass lymphoid tissue inducer (LTi) cells which promote both lymphoid organogenesis during development and the formation of intestinal lymphoid follicles in adults [2,7]. In respiratory tissue, NK cells and ILC1s protect from viral infection, ILC2s aid in the repair of damaged tissue caused by viral infections, and ILC3s block extracellular bacterial infections [2]. Although this review focuses on analyzing the major groups of ILCs (ILC1, ILC2, ILC3), it is beneficial to note the presence of a subpopulation of regulator ILCs (ILCregs). These ILCs are present within human and mouse intestinal mucosa, and upon stimulation via transforming growth factor β1 (TGF-β1), function to suppress intestinal inflammation via secretion of IL-10 to inhibit ILC1 and ILC3 activation [8].

Amongst respiratory tract infections, disease caused by *Mycobacterium tuberculosis* (*M. tb*) is the second leading cause of mortality worldwide, with an estimated 10.0 million cases and 1.5 million deaths each year [9]. The coordinated efforts of the innate and adaptive immune systems are crucial for adequate protection of a host from *M. tb* infection. Upon infection of *M. tb* via inhalation of aerosolized particles, antigen presenting cells (APCs), such as host alveolar macrophages and dendritic cells (DCs), mount the initial immune response by phagocytosing and destroying the bacilli. This occurs when APC receptors, such as toll-like receptors (TLRs), recognize pathogen associated molecular patterns (PAMPs) of *M. tb* [10]. Antigen presentation to T cells allows for their specialization into Th1 cells. The control of *M. tb* infection is driven primarily by the production of IFN-γ, secreted by CD4+ Th1 and NK cells under the stimulation of IL-12 and IL-18 [11]. IFN-γ functions to activate and recruit macrophages to the site of infection. These macrophages form an immune barrier around the mycobacteria, referred to as a granuloma, to suppress the infection and prevent bacterial dissemination. The resulting granuloma formation allows for the maintenance of latent *M. tb* infection. Following the formation of a granuloma, secretion of proinflammatory cytokines IL-1 and Il-6 leads to the recruitment of additional DCs, NK cells, and neutrophils for added control of the infection [10,12]. 

Although administration of the Bacille Calmette–Guérin (BCG) vaccine has been at the forefront of protection against tuberculosis (TB), the vaccine only provides partial protection which wanes over time [9,13]. However, along with BCG vaccination, recent studies have shown that ILCs provide a rapid pulmonary immune response against initial *M. tb* infection and play a crucial role in management. In this review, we will discuss the role of each group of major ILCs, in addition to NK cells, against *M. tb* infection, as well as how BCG vaccination may trigger early recruitment and drive effector functions of innate immune cells against *M. tb*.

## 2. Function of Natural Killer Cells against *M. tb*

NK cells are conventional ILCs and lymphocytic cells of the innate immune system that play an important role in the immune response against viruses, intracellular parasites, and intracellular bacteria [16]. They express CD16, an Fc receptor through which NK cells mediate ADCC, and CD56 antigenic marker [16,17]. In addition to these clusters of differentiation, NK cells also express activating receptors, termed NKp30, NKp44, and NKp46, the engagement of which leads to increased cytotoxicity [18,19]. 

The immune response against *M. tb* infection is dependent upon activation of CD4+ Th1 cells, CD8+ cytotoxic T cells, and NK cells rather than CD4+ Th2 cells [11,19]. Under the stimulation of IL-12 and IL-18 secreted by macrophages and DCs, NK cells constitute the early response to the bacilli by generating large quantities of IFN-γ, the most crucial cytokine necessary for immune defense against *M. tb* due to its ability to further stimulate phagocytosis and oxidative burst capacity within macrophages [20,21]. Dhiman et al., demonstrated that NK cells also enhance phagolysosome fusion within *M. tb* infected macrophages by producing IL-22, thereby further inhibiting intracellular *M. tb* growth [22]. In addition, NK cells carry out contact-induced apoptosis of mycobacteria infected cells, mediated by the release of perforin and granulysin. In a series of experiments conducted by Lu et al., the efficacy of primary human NK (pNK) cells against *M. tb* and *M. kansasii* were assessed under various modulations of perforin and granulysin levels to indicate the significance of NK cells during mycobacterial challenge. When pNK cells were incubated with both *M. tb* and *M. kansasii*, baseline levels of perforin and granulysin increased by 1.7- and 2.2-fold, respectively, indicating the enhancement of NK ADCC [23]. In subsequent experiments, depletion of intracellular cytotoxic granules by SrCl2, and depletion of mRNA levels of perforin and granulysin by siRNA resulted in a 2.5-fold and 3-fold decrease in *M. tb* and *M. kansasii* killing. Furthermore, when exogenous perforin and granulysin were added to cultures containing reduced concentration of perforin and granulysin, the efficacy of mycobacterial killing was restored [23]. To further demonstrate the importance of NK cells against *M. tb*, Vankayalapati et al. demonstrated that the depletion of CD56+ NK cells from peripheral blood mononuclear cells obtained from healthy tuberculin reactors not only resulted in markedly reduced IFN-γ levels, but also a reduction in the frequency of CD8+ IFN-γ+ T cells, resulting in a decreased capacity to destroy *M. tb*-infected macrophages [24].

Aside from their role in cytotoxicity, NK cells demonstrate the ability to readily recall effector responses to become more efficient during secondary exposure to haptens, viruses, and intracellular bacteria. This capacity of NK cells to mediate more efficient responses during secondary exposure to pathogens has been referred to as “innate memory” [16,25,26]. Fu et al. discovered that in patients with active tuberculosis, NK cells from pleural fluid cells expressed CD45RO, a marker that is also known to be expressed by primary antigen-specific memory T cells [27,28]. Using mean fluorescence intensity to analyze the expression of surface and intracellular molecules, Fu et al. found that CD45RO+ NK cells from pleural fluid cells expressed higher levels of IL-12Rβ1 and IL-12Rβ2, suggesting that this subset of NK cells was more sensitive to activation by IL-12 when compared with CD45RO- NK cells and thus exert a more robust production of IFN-γ for protection against *M. tb* [27,29].

## 3. Group 1 Helper ILCs against *M. tb*

The group 1 ILC subset includes both NK cells and non-cytotoxic ILC1s [5,30]. This classification is maintained by the fact that both NK cells and ILC1s are part of the innate immune system and have the capacity to mediate cytotoxic activity through the production of IFN-γ [31,32,33]. Although expression of both NK Cells and ILC1s depend on the T-bet transcription factor, expression of ILC1s can be distinguished from that of NK cells in that NK cells also require Eomesodermin [34]. 

As mentioned previously, *M. tb* infection within a host generates a type 1 immune response involving the production of large amounts of IFN-γ, mediated chiefly by the activation of CD4+ Th1 cells and NK cells in response to IL-12 and IL-18 [35]. In such an inflammatory environment, IFN-γ produced by ILC1s contributes to *M. tb* killing in host-infected macrophages. Furthermore, in a study conducted by Corral et al. within murine models, it was found that, under normal circumstances, respiratory tissue contains a small population of ILCs that express IL-18Rα, a phenotype signifying precursor ILCs with the ability to differentiate into mature ILC2 or ILC1-like cells. The inflammatory environment caused by *M. tb* infection leads to the differentiation of IL-18Rα+ ILCs into ILC1-like cells (characterized by the expression of T-bet), the functions of which resulted in decreased bacterial loads upon *M. tb* challenge [33]. In assays conducted by Pan et al., an analysis of blood samples collected from patients with active *M. tb* infection revealed a marked expansion of group 1 ILCs compared with samples from healthy patients [36]. This expansion of ILC1 population in blood samples with active tuberculosis also revealed increased production of IL-17, a proinflammatory cytokine, from ILC1s. Pan et al. also found that in patients with active *M. tb* infection, upregulation of DCs resulted in increased IL-23 secretion, driving greater differentiation of IL-17+ ILC1s relative to IFN-γ+ ILC1s [36]. As a result, in addition to IL-17 secretion from γδ T cells, secretion of IL-17 by group 1 ILCs further promotes granulopoiesis, neutrophil migration, and stimulates the innate immune response [36,37].

## 4. Group 2 Helper ILCs against *M. tb*

Group 2 ILCs are powerful effectors of type II inflammatory responses [38]. Although ILC2s are not abundant in the body, they are the most common type of ILC found within the lungs [39]. Regulation of ILC2s is crucial, as mishaps in the regulatory mechanisms can result in complications such as allergies and asthma [40]. ILC2s are typically involved in the immune response against helminth infection, tissue repair, and maintenance of metabolic homeostasis [41]. ILC2s work by releasing cytokines IL-5, IL-9, IL13, and epithelial growth factor amphiregulin, when stimulated by TSLP, IL-25, and IL-33 [2]. This results in a variety of physiological responses including eosinophil recruitment, stimulation of mucous production, and priming of alveolar macrophage [5,42].

Host immune response to *M. tb* relies on the functioning of cytokines associated with CD4+ Th1 cells, including IFN-γ, IL-2, and IL-12. These cytokines were discussed to be associated with the functioning of ILC1s, and function to fight intracellular bacteria and viruses. In contrast, Th2 cell-associated cytokines, IL-5, IL-4, and IL-13, function to fight extracellular pathogens. These correlate as cytokines released via ILC2 stimulation and have been found to inhibit the protective responses associated with Th1 cytokines [43].

Although expressed in intermediate levels, major histocompatibility complex (MHC) class II antigens allow ILC2s to act as a non-professional antigen presenting cells, thus recruiting CD4+ T cells in the immune response [44]. Murine studies conducted by Drake et al. found that lung ILC2 cells can work to enhance CD4+ T cells production, in addition to cytokine release [45]. In vitro proliferation of CD4+ T cells was between 1.4 and 2.8 times higher after 4 days when cultured with ILC2 cells, and a drastic increase was observed in IL-4, IL-5, and IL-13 concentrations [45]. Diedrich et al. studied IL-5 and IL-13 in the setting of coinfection of simian immunodeficiency virus (SIV) with *M. tb*. In doing so, they found that IL-5 may be decreasing *M. tb* mediated T cell response, and that antibody neutralization of IL-5 partially restored T cell response to the pathogen [46]. Similarly, Harris et al. found that IL-4 and IL-13 played a role in the inhibition on IFN-γ induced phagosome maturation within cells infected with *M. tb* variant *bovis* [47]. IL-13 and IL-4 decrease immunity against *M. tb* by inhibiting CD4+ Th1 mediated immunity, specifically suppressing macrophages that kill intracellular mycobacteria [47,48].

There still remains controversy regarding the perpetuating or protective role of ILC2s with regard to M. *tb* infection, as some studies suggest that ILC2s may be protective against bacterial infections. Yang et al. conducted an experiment to further study the phagocytic properties of pulmonary ILC2s in comparison properties of RAW264.7 macrophage cells. The results of the study suggested that although ILC2s are not as effective as RAW264.7 cells, they do have the ability to phagocytose bacteria [49]. Furthermore, similar to RAW264.7 cells, ILC2s were shown to have the capability of releasing extracellular trap formation in response to bacterial exposure. This suggests that ILC2s may have a more direct role in antibacterial processes [49]. Furthermore, Sudo et al. showed that ILC2s secrete granulocyte-macrophage colony-stimulating factor (GM–CSF), which is involved in the activation of macrophages that assist in fighting off *M. tb* infection [50,51].

## 5. Group 3 Helper ILCs against *M. tb*

The function and development of group 3 ILCs depend on the transcription factor retinoic acid receptor-related orphan receptor—γt (RORγt) [52]. The primary role of group 3 ILCs is to protect the mucosa against extracellular bacteria and fungi, by either secreting IL-17 alone or concurrently with IL-22 [53,54]. Ex vivo and murine experimentation by Fiancette et al. revealed that the loss of expression of RORγt completely reduced ILC3 secretion of IL-17, whereas IL-22 secretion was only reduced by about half [55]. In addition to providing mucosal protection, ILC3 and RORγt expression are required for the development of lymph nodes and Peyer’s patches in the intestine during embryogenesis [56]. When exposed to bacteria or fungi, ILC3s secrete IL-17 to activate antimicrobial peptide synthesis in epithelial cells, augmenting recruitment of neutrophils and macrophages to peripheral tissues [57]. 

Due to the pathogenesis of *M. tb* invading mucosal epithelial cells, ILC-3s play a role in the immune defense to fight off such infection and have been shown to mediate their protective role during the early phases of infection [58]. Upon host infection with *M. tb*, circulating levels of ILC3 in the bloodstream seem to decrease while the levels in the lungs increase [58]. By way of flow cytometry, Pan et al. calculated differences in ILC response in healthy human subjects compared with individuals infected by *M. tb*. The results revealed that individuals infected with *M. tb* had significantly increased levels of all ILC subtypes, including ILC3, and more importantly, increased IL-17 secretion without significant changes pertaining to IL-22 secretion [36]. As a result, the relationship between RORγt and ILC3 function indicates that RORγt expression enhances immune defense against *M. tb* infection [59]. 

Furthermore, it has been depicted that in tuberculosis infected patients, ILC3s have effectively upregulated 1438 genes. Amongst these genes, most are designated for the expression of proinflammatory cytokines and include CXCL1, CXCL5 and CXCL17, CCL2, which are necessary for the recruitment of neutrophils and macrophages, respectively [60,61]. As a result, it is evident that during *M. tb* infection, secretion of IL-17 by ILC3s subsequently increases macrophage and neutrophil chemotaxis to fight off the infection [57,62].

## 6. Effects of BCG Vaccine on Natural Killer Cells and Innate Lymphoid Cells

It is fairly known that the efficacy of the BCG vaccine is highly variable against adults with pulmonary TB, with estimates of protection that can range anywhere between 0 and 80%, whereas neonatal administration of BCG does provide protection against childhood manifestations such as TB-induced meningitis [63,64]. However, recent studies have shown that the administration of BCG can alter the effector functions of the innate immune system, providing the innate immunity with adaptive features [65]. 

In a series of animal and human trials conducted by Kleinnijenhuis et al., blood drawn from 29 patients before and after they were scheduled to receive the BCG vaccine was analyzed for cytokine measurements and for the role of NK cells [65]. Although peripheral blood analysis following BCG vaccination did not show a change in NK cell subsets, isolates of NK cells showed a slight increase in IFN-γ production both 2 weeks and 3 months following BCG vaccination, in addition to producing more proinflammatory cytokines for at least 3 months, particularly IL1β, when challenged with *M. tb* [65]. As mentioned previously, and as suggested by the literature, CD45RO+ memory-like NK from pleural fluid cells have the ability to produce stronger IFN-γ responses when compared with CD45RO- NK cells [27,66]. To assess whether administration of BCG vaccine altered effector functions of NK cells, Fu et al. discovered that in response to BCG and *M. tb* related antigens, CD45RO+ NK cells from pleural fluid cell not only produced two-to-three fold greater production of IL-22 compared to CD45RO- NK cells, but that they also produced higher levels of IL-12-dependent IFN-γ, suggesting that memory-like NK cells may display an exigent protective role against *M. tb* [27,66,67,68]. Recently, Suliman et al. also found for the first time that in a sample of healthy humans who were revaccinated with BCG, the frequency of peripheral blood BCG-reactive NK cells was boosted for at least one year after revaccination [69].

The effects of BCG vaccine administration on ILCs were further studied by Corral et al. in an experiment measuring ILC levels in mice provided with intranasal vaccine administration, compared with ILC levels in mice not given any form of BCG, 60 days prior to *M. tb* infection. Studies showed that 14 days post-infection, there was an increase in T-bet expression and ILC1-like cells isolated from the lungs of murine who were administered mucosal BCG vaccination, when compared with unvaccinated mice. As non-vaccinated mice 14 days post-infection showed an absence of IFN-γ producing ILC1-like cells, it is further suggested that the BCG vaccine instigated the increased ILC1-like cellular response [33]. Similarly, Steigler et al. conducted studies that supported the findings that intranasal BCG vaccination, more so than subcutaneous or intradermal, resulted in increased levels of ILC1 in both the lungs and lymph nodes of murine models [70]. 

Although Corral et al. found that BCG vaccination showed no advantageous post-infection increase in the levels of other ILCs apart from ILC1, other studies have found there to be an increase in ILCs in response to the BCG vaccination itself [33,70,71]. Steigler et al. showed that intranasal vaccination of mice resulted in increased levels of ILC1, ILC2, and ILC3 within lungs, 4 weeks post-immunization [41]. This is supported in future studies by Gunasena et al., showing increased levels of ILC1 and ILC3 in the tissue samples from lungs, liver, and spleen of mice 18 days post-immunization [71]. These studies have shown that BCG vaccination is involved with increased ILCs, yet there is room for further study regarding the interplay between BCG vaccination and its effects on ILCs post-infection with *M. tb*.

## 7. Conclusions

Infection with *M. tb* can have uncertainties in regard to disease outcome, whether the individual may develop active or latent Tuberculosis. Yet, one thing for certain is the robust immune response triggered by *M. tb* infection within immunocompetent individuals. The recent literature has demonstrated that ILCs play a critical role in bolstering the type I immune response against *M. tb* as they function to release cytokines and other inflammatory mediators necessary to fight infection. 

Group 1 ILCs, including NK cells, are involved in pathogen destruction and tissue restoration. Group 2 ILCs are found within the intestinal epithelium, respiratory tract, and are involved in fighting against helminths and viruses. Similarly, group 3 ILCs protect against invading pathogens and play a role in the development of lymphoid organs. In this literature review, we have discussed the details of these major ILC groups with attention to their role in *M. tb* infection and the effects of BCG vaccination on said ILCs. Our review highlights the role of ILCs within *M. tb* infection by detailing their pathogen-specific functions. In addition, our analysis of the literature pertaining to the effects of the BCG vaccine on ILCs provides insight on vaccination efficacy by highlighting its effects on each individual ILC involved in host immune response. Further studies can be done regarding the effect of BCG vaccine on ILC2 and ILC3 with respect to ILC levels post-infection with *M. tb*, to better assess long-term protection provided by BCG vaccination. A deeper understanding of the current BCG vaccine and role of ILCs on *M. tb* infection presents the possibility of creating a recombinant vaccine that includes the administration of ILCs to strengthen the host immune system.

## Figures and Tables

**Figure 1 biomedicines-10-02828-f001:**
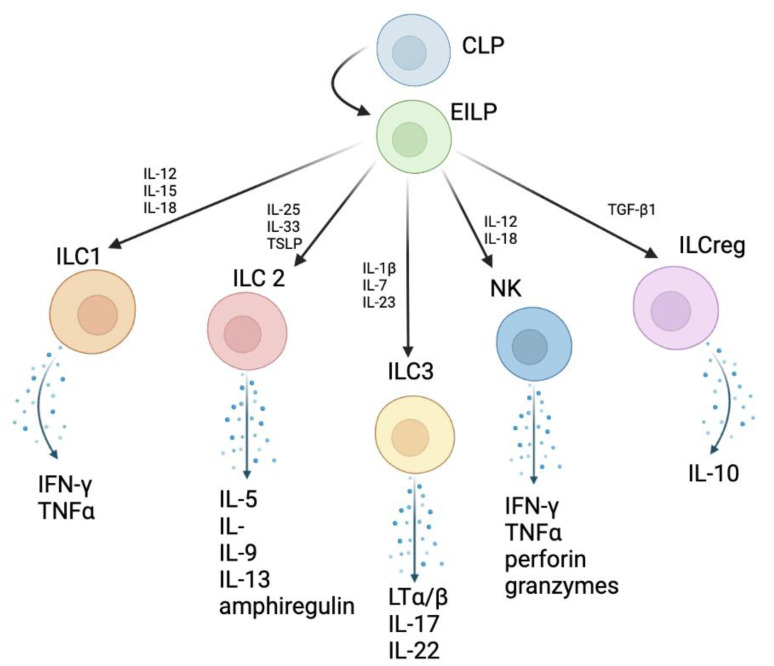
Distinction of each group of ILCs on the basis of cytokines stimulating their specialization and cytokines released by each ILC type. ILCs develop from common lymphoid progenitor cells (CLP) to early innate lymphoid progenitor cells (EILP), which are common and early innate lymphoid cells that are acted upon by varying cytokines to aid in the development of ILC1, ILC2, ILC3, ILCregs, and NK cells. Now specialized, these cells will release cytokines involved in inflammatory processes. The major cytokines involved in these processes are listed above [14,15].

## Data Availability

Not applicable.

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
