# Peer review of "Immunologic Role of Innate Lymphoid Cells against Mycobacterial tuberculosis Infection"

_biomedicines, 2022, doi:10.3390/biomedicines10112828_

Round 1

Reviewer 1 Report

This review is focused and succinctly informative of the subject of ILCs in tuberculosis, a group of cells currently gaining more attention both in infection and inflammatory diseases.   It is very well-written, focused on the specific topic of ILCs in tuberculosis, and structured beautifully with ILC definitions and the role of each in the pathology and vaccine-induced prevention/therapy (all appropriately supported via literature).  While making concerted attempts to find areas for improvements (such as inaccuracies or striking omissions of literature), I was unsuccessful.  Therefore, I find this review to be acceptable for publication in its current condition.

Author Response

Dear Reviewer#1,

We appreciate the time and effort you have all taken to provide an extensive and thorough review of our manuscript. We are also very encouraged by how well it was accepted, and we agree that the suggested edits have greatly improved our manuscript. The objective of our paper was to provide a brief discussion about the immune defense roles of the various innate lymphoid cells against M. tb infections. We hope that our revised manuscript based on all of the suggested edits make this objective much clearer.

Thank you all for you valuable time.

Reviewer #1:

This review is focused and succinctly informative of the subject of ILCs in tuberculosis, a group of cells currently gaining more attention both in infection and inflammatory diseases.   It is very well-written, focused on the specific topic of ILCs in tuberculosis, and structured beautifully with ILC definitions and the role of each in the pathology and vaccine-induced prevention/therapy (all appropriately supported via literature).  While making concerted attempts to find areas for improvements (such as inaccuracies or striking omissions of literature), I was unsuccessful.  Therefore, I find this review to be acceptable for publication in its current condition.

  • Greatly appreciate you for taking the time to read our manuscript and providing such great feedback with your acceptance. Thank you!

Reviewer 2 Report

In this review, the authors make a state of art of the role of Innate Lymphoid cells against Mycobacterial tuberculosis infection. The manuscript is well-written and organized. However, there are some minor points to be modified and suggestions for improving the text. Please see specific comments below.

1) In a separate paragraph, a brief overview of immune defence against Mtb. could be added.

2) Some information about a newly identified member — ILCregs that share immunosuppressive features with Tregs could be added.

Author Response

Dear Reviewer#2,

We appreciate the time and effort you have all taken to provide an extensive and thorough review of our manuscript. We are also very encouraged by how well it was accepted, and we agree that the suggested edits have greatly improved our manuscript. The objective of our paper was to provide a brief discussion about the immune defense roles of the various innate lymphoid cells against M. tb infections. We hope that our revised manuscript based on all of the suggested edits make this objective much clearer.

Thank you all for you valuable time.

Reviewer #2:

In this review, the authors make a state of art of the role of Innate Lymphoid cells against Mycobacterial tuberculosis infection. The manuscript is well-written and organized. However, there are some minor points to be modified and suggestions for improving the text. Please see specific comments below.

  • In a separate paragraph, a brief overview of immune defense against Mtb. could be added.
  • Thank you for this suggestion! Within the introduction, we attempted to discuss the immune defense against M. tb; however, we have elaborated on the paragraph pertaining to this point and hope it makes things clearer. (Lines 64-80)
  • Some information about a newly identified member — ILCregs that share immunosuppressive features with Tregs could be added.
  • Interesting suggestion! We found some Information regarding the subpopulation of regulatory ILCs (ILCregs) and have added to the introduction (Lines 59-63) Thank you! We have also incorporated ILCregs into our figure 1 as well.

Reviewer 3 Report

In this review, the authors discussed the mechanistic role of innate lymphoid cells in case of Mycobacterium tuberculosis infection, and the effects of the BCG vaccine on innate lymphoid cells. The review is interesting and concise. I have only noticed that in the abstract: line 14: the authors mentioned that innate lymphoid cells participate "against" the protective immune response to boost the type I immune response!!!! Please revise this sentence.

Additionally, moderate revision of the language and the abbreviations is required.

Author Response

Dear Reviewer#3,

We appreciate the time and effort you have all taken to provide an extensive and thorough review of our manuscript. We are also very encouraged by how well it was accepted, and we agree that the suggested edits have greatly improved our manuscript. The objective of our paper was to provide a brief discussion about the immune defense roles of the various innate lymphoid cells against M. tb infections. We hope that our revised manuscript based on all of the suggested edits make this objective much clearer.

Thank you all for you valuable time.

Reviewer #3:

In this review, the authors discussed the mechanistic role of innate lymphoid cells in case of Mycobacterium tuberculosis infection, and the effects of the BCG vaccine on innate lymphoid cells. The review is interesting and concise. I have only noticed that in the abstract: line 14: the authors mentioned that innate lymphoid cells participate "against" the protective immune response to boost the type I immune response!!!! Please revise this sentence.

Additionally, moderate revision of the language and the abbreviations is required.

  • Excellent catch, the sentence has been corrected and revised to be more clear - “Recent discoveries about innate lymphoid cells (ILCs) have provided further insight about how these cells participate within the protective immune response against M. tb infection and help boost the type I immune response.”

In addition, the manuscript has been reviewed for language, grammar and abbreviation mistakes. Appropriate revisions have been made.